Diversity and distribution of Symbiodiniaceae detected on coral reefs of Lombok, Indonesia using environmental DNA metabarcoding

Pratomo Arief 1 2
Bengen Dietriech G. 2
Zamani Neviaty P. 2
Lane Christopher 3
Humphries Austin T. 4
Borbee Erin 3
Subhan Beginer beginersubhan@apps.ipb.ac.id 2
Madduppa Hawis 2 5
1 Raja Ali Haji Maritime University , Tanjungpinang , Indonesia
2 Department of Marine Science and Technology, Institut Pertanian Bogor , Bogor , Indonesia
3 Department of Biological Sciences, University of Rhode Island , Rhode Island , United States of America
4 Department of Fisheries, Animal and Veterinary Sciences, University of Rhode Island , Rhode Island , United States of America
5 Oceanogen Research Center , Bogor , Indonesia
Pochon Xavier
Electronic publication date: 2022 Oct 24
Publication date: 2022
Volume: 10
Electronic Location ID: e14006
Received 2021 Oct 6; Accepted 2022 Aug 14
Copyright: ©2022 Pratomo et al.
Copyright year: 2022
Copyright holder: Pratomo et al.
License: This is an open access article distributed under the terms of the Creative Commons Attribution License, which permits unrestricted use, distribution, reproduction and adaptation in any medium and for any purpose provided that it is properly attributed. For attribution, the original author(s), title, publication source (PeerJ) and either DOI or URL of the article must be cited.
License URL: https://creativecommons.org/licenses/by/4.0/

Keywords: Coral triangle, Coral hosts, Endosymbiotic dinoflagellate, Aquatic plankton, Benthic periphyton, Next generation biomonitoring

Funding: BPPDN scholarship and USAID through the Sustainable Higher Education Research Alliances (SHERA) Program—Centre for Collaborative Research on Animal Biotechnology and Coral Reef Fisheries (CCR ANBIOCORE) of IPB University This research was supported by the BPPDN scholarship and USAID through the Sustainable Higher Education Research Alliances (SHERA) Program—Centre for Collaborative Research on Animal Biotechnology and Coral Reef Fisheries (CCR ANBIOCORE) of IPB University. The funders had no role in study design, data collection and analysis, decision to publish, or preparation of the manuscript.

==============================
Background

Dinoflagellates of family Symbiodiniaceae are important to coral reef ecosystems because of their contribution to coral health and growth; however, only a few studies have investigated the function and distribution of Symbiodiniaceae in Indonesia. Understanding the distribution of different kinds of Symbiodiniaceae can improve forecasting of future responses of various coral reef systems to climate change. This study aimed to determine the diversity of Symbiodiniaceae around Lombok using environmental DNA (eDNA).

Methods

Seawater and sediment samples were collected from 18 locations and filtered to obtain fractions of 0.4–12 and >12 µm. After extraction, molecular barcoding polymerase chain reaction was conducted to amplify the primary V9-SSU 18S rRNA gene, followed by sequencing (Illumina MiSeq). BLAST, Naïve-fit-Bayes, and maximum likelihood routines were used for classification and phylogenetic reconstruction. We compared results across sampling sites, sample types (seawater/sediment), and filter pore sizes (fraction).

Results

Phylogenetic analyses resolved the amplicon sequence variants into 16 subclades comprising six Symbiodiniaceae genera (or genera-equivalent clades) as follows: Symbiodinium, Breviolum, Cladocopium, Durusdinium, Foraminifera Clade G, and Halluxium. Comparative analyses showed that the three distinct lineages within Cladocopium, Durusdinium, and Foraminifera Clade G were the most common. Most of the recovered sequences appeared to be distinctive of different sampling locations, supporting the possibility that eDNA may resolve regional and local differences among Symbiodiniaceae genera and species.

Conclusions

eDNA surveys offer a rapid proxy for evaluating Symbiodiniaceae species on coral reefs and are a potentially useful approach to revealing diversity and relative ecological dominance of certain Symbiodiniaceae organisms. Moreover, Symbiodiniaceae eDNA analysis shows potential in monitoring the local and regional stability of coral–algal mutualisms.

Introduction

Symbiodiniaceae, also known as zooxanthellae, play vital roles within their coral hosts, such as providing energy, absorbing residual metabolites, and promoting growth (Davy, Allemand & Weis, 2012; Purnomo, 2014). These symbionts also contribute to the adaptability and resilience of corals to environmental change, especially ocean warming (Berkelmans & Van Oppen, 2006; Baskett, Gaines & Nisbet, 2009; Suggett, Warner & Leggat, 2017; Claar et al., 2020; Howells et al., 2021). Stress-tolerant Symbiodiniaceae can improve the survival of coral colonies exposed to thermal stress (Abrego et al., 2008; LaJeunesse et al., 2010a; LaJeunesse et al., 2010b; Stat & Gates, 2011; Cunning, Silverstein & Baker, 2015; Bourne, Morrow & Webster, 2016; Hoadley et al., 2019). Therefore, understanding the potential diversity of Symbiodiniaceae is necessary in forecasting the future of coral reef ecosystems in different regions under a rapidly changing climate.

Endosymbiotic dinoflagellates of family Symbiodiniaceae are extremely prevalent in coral reef ecosystems. Symbiodiniaceae engage in mutualistic relationships with various invertebrates, including scleractinian corals, octocorals, anemones, jellyfishes, mollusks, sponges, flatworms, and foraminifera (Pochon et al., 2001; LaJeunesse et al., 2010a; LaJeunesse et al., 2010b; LaJeunesse et al., 2018; Pochon, Putnam & Gates, 2014). A number of Symbiodiniaceae live as aquatic plankton and benthic periphyton, and some are associated with macroalgae and seagrasses (Venera-Ponton et al., 2010; Takabayashi et al., 2012; Fujise et al., 2021). To date, 11 named genera have been classified as members of the Symbiodiniaceae family: Symbiodinium (clade A), Philozoon (temperate clade A), Breviolum (clade B), Cladocopium (clade C), Durusdinium (clade D), Miliolidium (foraminifera clade D), Effrenium (clade E), Freudhentalidium (clade Fr3), Fugacium (clade Fr5), Gerakladium (clade G), and Halluxium (clade H) (LaJeunesse et al., 2018; LaJeunesse et al., 2021; Nitschke et al., 2020; Pochon & LaJeunesse, 2021). However, there are 16 distinct lineages, with Foraminifera Clade G, Clade Fr2, Clade Fr4, Clade I, and Clade J representing the undescribed genera (LaJeunesse et al., 2018; Yorifuji et al., 2021).

Indonesia is a part of the Coral Triangle (Veron et al., 2009; Gelis et al., 2021), and coral reef ecosystems are a valuable economic resource for coastal communities across the archipelago; however, data on the diversity of Indonesian Symbiodiniaceae are still limited (Loh, Cowlishaw & Wilson, 2006; Bo et al., 2011; Purnomo, 2014; DeBoer et al., 2012). Previous studies about Symbiodiniaceae from areas in the region such as the South China Sea, Thailand, Singapore, Palau, the Philippines, and Timor-Leste, only focused on Symbiodiniaceae populations within their host organisms. Some of the Symbiodiniaceae genera in these reports include Symbiodinium, Breviolum, Cladocopium, Durusdinium, Gerakladium, and Fugacium (Fabricius et al., 2004; Loh, Cowlishaw & Wilson, 2006; Reimer & Todd, 2009; LaJeunesse et al., 2010a; LaJeunesse et al., 2010b; Taguba, Sotto & Geraldino, 2016; Tong et al., 2018; Brian, Davy & Wilkinson, 2019). However, little is known about Symbiodiniaceae within Indonesian waters, which is the most biodiverse mairne region in the world.

Symbiodiniaceae cannot be directly identified using conventional microscopy. The need for collection and isolation from multiple locations increases the difficulty of assessing this taxonomic group. Advances in the use of environmental DNA (eDNA) and multitaxon sequencing techniques (metabarcoding) have allowed the study of Symbiodiniaceae communities through the collection of environmental samples, such as water and sediment (Arif et al., 2014; Shinzato et al., 2018; Fujise et al., 2021). The advantages of the eDNA-based approach include ease of use, noninvasive nature, broad spatial scale, and cost effectiveness (Deiner et al., 2017).

This study aimed to develop a rapid proxy for estimating the diversity of Symbiodiniaceae in water and sediment samples from the coral reef ecosystems around Lombok Island in Indonesia using eDNA. We make comparisons across sampling sites, eDNA source (seawater/sediment), and filter pore sizes (fraction). A better understanding of the diversity and composition of Symbiodiniaceae in Indonesian coral reefs is important for conservation and management of marine ecosystems.

Methods

Study sites

This study was conducted in coral reef habitats around Lombok Island, West Nusa Tenggara Province, Indonesia. This island is the constituent of the marine ecoregion of Nusa Tenggara (Lesser Sunda), which has a coral reef area of about 272,123 ha (Giyanto Abrar et al., 2017). The western part of Lombok Island is directly adjacent to Lombok Strait and its southern part is the Indian Ocean. The study areas were located 5–100 m from shore, with depths ranging 1–10 m and a mean tidal range of about 1.8 m. Samples were collected from 5th to 12th July 2018 (Table 1).

Table 1 Coordinates of sampling stations at each coastal area around Lombok Island, Indonesia.

Coastal area	Station	Date	Depth (m)a	Position	
				South	East	
East Lombok	Gili Sulat 01	5 July 2018	<1	08°19.069′	116°42.355′	
	Gili Lawang	6 July 2018	1.2	08°17.833′	116°41.290′	
	Gili Sulat 02	5 July 2018	>10	08°18.900′	116°43.519′	
	Gili Sulat 03	5 July 2018	<1	08°18.574′	116°42.767′	
	Gili Petagan	6 July 2018	2.8	08°24.698′	116°45.324′	
	Gili Kondo	6 July 2018	<1	08°26.572′	116°44.016′	
North Lombok	Gili Trawangan 01	11 July 2018	8.46	08°21.253′	116°01.505′	
	Gili Air	12 July 2018	<1	08°21.854′	116°04.369′	
	Gili Trawangan 02	11 July 2018	1.4	08°20.271′	116°02.280′	
	Gili Meno	11 July 2018	>10	08°20.852′	116°03.077′	
	Tanjung Sire 01	12 July 2018	4.8	08°21.455′	116°06.506′	
	Tanjung Sire 02	12 July 2018	8.3	08°22.001′	116°05.840′	
West Lombok	Gili Nanggu	8 July 2018	<1	08°42.887′	116°00.362′	
	Gili Rengit	9 July 2018	<1	08°43.114′	115°55.135′	
	Gili Golek	9 July 2018	<1	08°44.967′	115°53.405′	
	Gili Gede	9 July 2018	<1	08°44.045′	115°54.945′	
	Tanjung Bunutan 01	8 July 2018	>10	08°43.693′	116°02.848′	
	Tanjung Bunutan 02	8 July 2018	>10	08°43.039′	116°02.363′	
Notes.

a In lowest low water level (LLWL) based on Hydrographic and Oceanographic Center, The Indonesian Navy (2007) and mean tidal range is 187 cm.

eDNA sample collection

During the survey, eDNA seawater and sediment samples were collected by scuba diving from six reef stations within each coastal area (West Lombok, East Lombok, and North Lombok). Two samples (one seawater and one sediment) per station were collected per day from three stations, in total, 72 samples in Lombok (Fig. 1 and Tables 1 and 2). The distance between the sampling stations was at least 1500 m to avoid overlap. At each station, 4 L of seawater was collected from the water column (∼2 m above the reef substrate) as well as a sediment sample (water + sediment in 1:1 ratio) in sterilized bottles. Before sampling, the bottle was rinsed with a 30% commercial bleach solution, followed by distilled water. The collected eDNA samples were stored in a cool box and brought to basecamp at Lombok Island as soon as possible (less than 12 h). Each sample was filtered twice using a peristaltic pump (Thermo Fisher Scientific) through 47 mm diameter polycarbonate membrane filters (Sterlitech) with two different pore sizes: 12 µm first and then 0.4 µm. According to Turner et al. (2014), a combination of ≥ 0.2 µm filtration pore size and water volume enables optimal eDNA capture and maximize detection probability. In addition, a large pore size is required to avoid clogging the filters. The sediment samples, were shaken first and then filtered 1–2 min after shaking. Each filter was cut into two, and each half was placed in a 1.5 mL vial prefilled with DNA Shield as a preservative. At the end of all eDNA survey activities, all the samples were transported to the Marine Biodiversity and Biosystematics Laboratory at Bogor Agricultural (IPB) University, Indonesia, via commercial courier and then stored at −20 °C until DNA extraction.

Figure 1 Map of the research sites around Lombok Island, Indonesia.

(A) West Lombok, (B) North Lombok, and (C) East Lombok.

Table 2 Successfully amplified eDNA samples by sample type and filter pore size.

EB356–EB396 are the sample codes; n.a. (not available) indicates the eDNA samples were not successfully amplified; bold font indicates Symbiodiniaceae were detected.

Location	Station	Seawater fraction	Sediment fraction	
		0.4–12 µm	>12 µm	0.4–12 µm	>12 µm	
East Lombok	Gili Sulat 1	n.a.	EB356	EB357	EB358	
	Gili Lawang	EB367	EB368	EB369	EB370	
	Gili Sulat 2	EB359	EB360	EB361	EB362	
	Gili Sulat 3	EB363	EB364	EB365	EB366	
	Gili Petagan	n.a.	EB371	EB372	EB373	
	Gili Kondo	n.a.	EB374	EB375	EB376	
West Lombok	Gili Nanggu	n.a.	n.a.	EB377	n.a.	
	Gili Rengit	n.a.	n.a.	EB379	n.a.	
	Gili Golek	n.a.	n.a.	EB380	EB381	
	Gili Gede	EB382	n.a.	EB383	n.a.	
	Bunutan 1	n.a.	n.a.	EB378	n.a.	
	Bunutan 2	n.a.	n.a.	n.a.	n.a.	
North Lombok	Gili Trawangan 1	EB384	EB385	EB386	EB387	
	Gili Air	EB396	n.a.	n.a.	n.a.	
	Gili Trawangan 2	EB388	EB389	EB390	EB391	
	Gili Meno	EB392	EB393	EB394	EB395	
	Tanjung Sire 1	n.a.	n.a.	n.a.	n.a.	
	Tanjung Sire 2	n.a.	n.a.	n.a.	n.a.	

eDNA seawater sampling in this study was permitted within the framework of the United States Agency for International Development—Sustainable Higher Education Research Alliances (USAID-SHERA) program through the Centre for Collaborative Research Animal Biotechnology and Coral Reef Fisheries of IPB University, award no. AID-497-A-16-00004. The field research permit was issued by IPB University Rector (Surat Tugas no. 403/IT3/KP/2019). Permits for this research were issued by the Indonesian Ministry of Research and Technology to EB (130/E5/E5.4/SIP/2019), CL (461/SIP/FRP/E5/Dit.KI/XII/2017), and AH (455/SIP/FRP/E5/Dit.KI/XII/2017).

DNA extraction, amplification, and sequencing

The filtered eDNA samples were extracted and amplified at the Marine Biodiversity and Systematic Laboratory of IPB University and sequenced at the University of Rhode Island (URI) Genomics and Sequencing Center, United States of America. DNA was extracted from the filters using ZymoBiomics Miniprep Kit (Zymo Research, Irvine, CA, USA) following the manufacturer’s instructions. V9 hypervariable regions of the eukaryotic small sub unit (SSU) 18S ribosomal RNA (rRNA) genes were amplified using a polymerase chain reaction (PCR) platform and prepared for 2 × 250 bp paired-end Illumina MiSeq sequencing (Illumina, San Diego, CA, United States). Amplification was conducted using V9 primer set 1389F: 5′-TTG TAC ACA CCG CCC-3′ and 1510R: 5′-CCT TCY GCA GGT TCA CCT AC-3′ (Amaral-Zettler et al., 2009; Stoeck et al., 2010), Illumina adapters, linker sequences, index, and pad (Kozich et al., 2013). The PCR profile used was as follows: 3 min at 94 °C, followed by 35 cycles of 94 °C for 45 s, 48 °C for 30 s, and 72 °C for 30 s, and a final extension at 72 °C for 5 min. Each 49 µL of PCR reaction comprised 25 µL of MyTM HS red mix (Bioline Ltd., London, UK), 1 µL of (10 µM) forward primer, 1 µL of (10 µM) reverse primer, and 1 µL of DNA template. The final volume was adjusted to 49 µL using ddH2O. 1x reaction was 0.2 µM. The PCR product was checked via the electrophoresis final master mix concentration in 1 × reaction was 0.8 ×, and the final primer concentration in of 5 µL of aliquots on 1% agarose gel in 0.5X TBE buffer. Library preparation and sequencing were performed at URI. A second PCR was performed to add the dual indices and Illumina sequencing adapters from the TruSeq PCR-Free LT kit to the target amplicons, using Kapa HotStart HiFi 2x ReadyMix DNA polymerase (Kapa Biosystems Ltd., London UK). The PCR profile used was as follows: initial denaturation at 95 °C for 3 min, followed by 9 cycles of 95 °C for 30 s and 55 °C for 30 s, and final extension at 72 °C for 5 min. The presence and length (bp) of the PCR product or amplicon were tested by electrophoresis. Successful amplicons were then purified using paramagnetic Kapa pure beads (bead-to-sample volumetric ratio in 1.6:1). A Qubit fluorometer with Qubit dsDNA HS Assay reagent (Invitrogen, California, US) was used to quantify all libraries. The prepared samples were combined in equal concentrations and then pooled with a 20% denatured and diluted PhiX Illumina control library. The final pooled library was sequenced on an Illumina MiSeq with the MiSeq v2 500-cycle kit (Illumina, San Diego, CA, United States). After quality checking, only 41 out of 72 samples were found to be of sufficiently high quality for sequencing (Table 2). The low quality of some libraries may be due to eDNA degradation during sample transport and extraction.

Data processing and bioinformatic analyses

The obtained forward and reverse raw sequence data were converted to demultiplexed fastq files (see additional information on data availability). The sequence read quality was checked using FastQC v.0.11.8 (https://www.bioinformatics.babraham.ac.uk) at each analysis step. Cutadapt v.1.18 (Martin, 2011) was used to trim the reverse and forward primer sequences and remove short reads with lengths <100 bp and low quality reads with a Phred Q score of <20. Qiime2.2019.10 pipeline (Caporaso et al., 2010; Bolyen et al., 2019) was employed for further data processing. DADA2 v.2018.11.0 (Callahan et al., 2016) (via q2-dada2) was applied for denoising, joining denoised paired-end reads, filtering out chimeric sequences and singletons, and dereplicating sequences to produce amplicon sequence variants (ASVs). Owing to the high quality of the sequences obtained after Cutadapt procedure, trimming and truncating were not performed during DADA2 processing.

ASV identification

Symbiodiniaceae species were identified from the eDNA sequences by classifying all ASVs (File S1) using the q2-feature-classifier (Bokulich et al., 2018) classify-sklearn Fit-Naïve Bayes taxonomy classifier against the 18S NR SILVA (release 123 Qiime compatible) 97% and 99% OTU reference sequences (https://www.arb-silva.de/download/archive/qiime/). Stoeck et al. (2010) showed the differential increase in diversity detected when the V9 dataset is clustered at 97%, 98%, 99%, and 100% sequence similarity for the minimum expected error rate. Putative Symbiodiniaceae ASVs were then filtered from the obtained eukaryote taxonomy table (File S2) (Table 3) and then assessed using the NCBI BLAST routine by selecting the best hit at >95% identity in the nr/nt database of NCBI (https://www.ncbi.nlm.nih.gov/, accessed on 1/19/2020, version 2.11.0). The BLAST results (File S3) were evaluated, and reference sequences (accessions) were selected for further analyses. Additional SSU 18S Symbiodiniaceae reference sequences (accessions) representing several families in the order Suessiales, family Symbiodiniaceae were obtained from the NCBI database and the V9-SSU 18S sequence reference database of TARA Ocean Expedition (Decelle et al., 2018) and Loh, Cowlishaw & Wilson (2006). The final compiled reference sequence database (File S4) contained 82 sequences. These reference sequences and the putative Symbiodiniaceae ASVs from the samples were then aligned with MAFFT v.7 (Katoh & Standley, 2013) (via q2-alignment), followed by masking (Rajan, 2012). A phylogenetic tree representing the evolutionary relationships of Symbiodiniaceae members was constructed using the maximum likelihood approach in the IQ-TREE v.1.6.12 (Nguyen et al., 2015) (via q2-phylogeny) with 1,000 bootstraps. These parameters were adopted to calculate the phylogenetic branch support scores from Shimodaira and Hasegawa approximate likelihood ratio test (SH-alrt) with local bootstraps (lbt), Bayesian (abayes), and ultrafast bootstraps (ufboot). Detailed explanations for these scores are provided in the IQ-TREE documentation (Minh et al., 2021). The best-fit substitution model TIM3 + F + R3 was chosen according to the Bayesian Information Criterion by ModelFinder applied in IQ-TREE (Kalyaanamoorthy et al., 2017). The Symbiodiniaceae taxonomic nomenclature was adopted from LaJeunesse et al. (2018). The term subclade was used instead of species because the 18S short eDNA sequence cannot be resolved to species-level for Symbiodiniaceae.

Table 3 Summary of Symbiodiniaceae classifications.

ASVs were classified using a probabilistic Bayesian method referring to SILVA database at similarities of 97% and 99%, NCBI database BLAST routine, and phylogenetic reconstruction.

Methods →	Fit-Classifier-Naïve Bayes	BLAST	Phylogenetics	
DB Ref.*→	SILVA (97&99%)	Conf. rates**	Accession no.	NCBI d	%Id.***	Genera/subclades	Scores: SH-alrt/lbt/ abayes/ufboot	
OTUs								
OTU.sym1	Symbiodinium	0 .999410455	KC816641.1	Cladocopium sp. clade C	100	Cladocopium/C.sym1	93.2/90.8/1/69	
OTU.sym2	Symbiodinium	0.823380237	AY165766.1	Symbiodinium sp. ex P. briareum/D1b	100	Durusdinium/D1.sym2	84.5/79.8/0.875/62	
OTU.sym3	Symbiodinium	0.890746449	EF526860.1	uncultured marine Eukaryote	99.24	Unclassified Suessiaceae/OTU.sym3	100/100/1/100	
OTU.sym4	Symbiodinium	0.78273645	MH702366	CladeG2_V9_7221c		Formaninifera Clade G/. G2.sym4	54.8/64.4/0.838/63	
OTU.sym5	Symbiodinium	0.768964536	AB085912.1	Cladocopium sp.c		Cladocopium/ C.sym5	93.2/90.8/1/69	
OTU.sym6	Symbiodinium	0.993577182	AF238261.1	Symbiodinium sp. clade E/D1b	99.24	Durusdinium/ D1.sym6	84.5/79.8/0.875/62	
OTU.sym7	Symbiodinium	0.995164623	KC816641.1	Cladocopium sp. clade C	99.24	Cladocopium/C.sym7	93.2/90.8/1/69	
OTU.sym8	Symbiodinium	0.999188819	AF238258.1	Symbiodinium sp. type C	99.24	Cladocopium/C.sym8	93.2/90.8/1/69	
OTU.sym9	Symbiodinium	0.82770918	KP404862.1	uncultured Eukaryotec		Unclassified Suessiaceae/OTU.sym9	94.9/96.6/1/91	
OTU.sym10	Symbiodinium	0.925078226	AB085912.1	Cladocopium sp.c		Cladocopium/C.sym10	93.2/90.8/1/69	
OTU.sym11	Symbiodiniuma	0.998952834	LN898222.1	Yihiella yeosuensis	100	Yihiella/OTU.sym11	83.5/80.5/0.963/79	
OTU.sym12	Symbiodinium	0.991677932	MH702343.1	Symbiodinium sp. clade H	97.71	Halluxium/H.sym12	89.2/87/0.964/83	
OTU.sym13	Symbiodiniuma	0.942758624	LN898222.1	Yihiella yeosuensis	99.24	Yihiella/OTU.sym13	83.5/80.5/0.963/79	
OTU.sym14	Symbiodinium	0.890990315		Incertae Sedis		Unclassified Suessiaceae/OTU.sym14	68/62.4/0.43/91	
OTU.sym15	Symbiodinium	0.999046044	KC816642.1	Cladocopium sp. clade C	99.24	Cladocopium/C.sym15	93.2/90.8/1/69	
OTU.sym16	Symbiodinium	0.999527293	HM067612.1	Symbiodinium sp. 2-125/CladeCb	99.24	Cladocopium/C.sym16	93.2/90.8/1/69	
OTU.sym17	Symbiodinium	0.987978975	MMETSP1371	Symbiodinium C15c		Cladocopium/C.sym17	93.2/90.8/1/69	
OTU.sym18	Symbiodinium	0.997997906	LK934670.1	Breviolum minutum	99.24	Breviolum/B.sym18	100/100/1/99	
OTU.sym19	Symbiodinium	0.994027696	AF238261.1	Symbiodinium sp. clade E/D1b	99.24	Durusdinium/D1.sym19	84.5/79.8/0.875/62	
OTU.sym20	Symbiodinium	0.742593275	LC361448.1	Ansanella natalensis	98.47	Ansanella/OTU.sym20	26.8/54.4/0.465/52	
OTU.sym21	Symbiodinium	0.978735399	AB085913.1	Cladocopium sp.c		Symbiodinium/A.sym21	100/100/1/100	
OTU.sym22	Symbiodinium	0.977772371	AY165766.1	Symbiodinium sp. ex P. briareum/D1b	100	Durusdinium/D1.sym22	84.5/79.8/0.875/62	
Notes.

a In SILVA 97%, OTU.sym13 was classified as Symbiodinium, but in SILVA 99%, OTU.sym13 and OTU.sym11 were classified as Polarella. Therefore, the further analysis considered Polarella as possibly belonging to the Symbiodiniaceae.

b According to Decelle et al. (2018).

c Non BLAST result.

d Nearest subclade branch (see Fig. 3).

* Reference database.

** Confidence level.

*** Percentage identity.

Statistical analyses

The relative abundance data for the putative Symbiodiniaceae taxa (File S5) were from DADA2 results and used as input for Venn diagram and statistical analyses. Venn diagram analyses were performed using the online application in http://bioinformatics.psb.ugent.be/webtools/Venn/ to compare the Symbiodiniaceae individuals across different locations (coastal area), eDNA source (seawater and sediment samples), and fractions (filter pore size). We determined the most commonly distributed subclades and distinctive subclades to each location/station. Statistical analyses were used to compare Symbiodiniaceae abundance, diversity, and features observed across different sites, sample types, and fractions. All these statistical analyses were carried out on Qiime2.2019.10 pipeline (Caporaso et al., 2010; Bolyen et al., 2019). Alpha diversity (observed features and Shannon’s entropy) and beta diversity (Bray–Curtis dissimilarity) were estimated using q2-diversity after the samples were rarefied (subsampled without replacement) to 28 sequences per sample. The comparison of all samples were grouped by location, eDNA source, and fraction to examine differences in abundance and alpha diversity employing the Kruskal–Wallis test (Kruskal & Wallis, 1952) and beta diversity applying the Permanova test (Anderson, 2001) using 9999 permutations.

Results

Obtained sequences, ASVs, and eukaryote classification

From the 72 samples across 18 stations, DNA was successfully extracted from 41 samples at 16 stations, yielding a total of 3,168,655 raw sequences and about 30,205–240,604 sequences per sample (Table 2 and Fig. S1). DADA2 yielded a total of 20,486 ASVs (File S1). The mean length of the obtained sequences was 127.81 ± 22.03 bp. The ASV classification demonstrated the potential diversity of eukaryotes in the reef waters of Lombok Island. According to the total ASVs classified to taxon level 4 from the SILVA database, the dominant taxon was unclassified eukaryotes (43.35%), followed by Metazoa (9.47%), Ochrophyta (7.83%), Dinoflagellates (4.5%), and Discicristata (4.4%) (Fig. 2).

Figure 2 Proportion of Eukaryote taxa.

Based on the total ASVs of taxon level 4 out of 15 taxon levels according to the SILVA database (https://www.arb-silva.de/).

Symbiodiniaceae detection and classification

Table 3 summarizes the results of Symbiodiniaceae classification performed using a Eukaryote classifier (File S2) and BLAST (File S3) and phylogenetic analyses. The probabilistic classifier detected and classified the Symbiodiniaceae taxa at the family level. Twenty-two ASVs (named OTU.sym1 to OTU.sym22) were found to be putative Symbiodiniaceae with confidence levels ranging 0.743–0.999 (Table 3). BLAST results indicated that some ASVs were neither Symbiodiniaceae nor classified at the genus level. A partial phylogenetic reconstruction of the families in order Suessiales was conducted using the reference sequences obtained from the searched databases (File S4) and the putative Symbiodiniaceae ASV sequences from the study (Fig. 3A). Only 16 out of the 22 ASVs were identified as members of the monophyletic group of the Symbiodiniaceae family clade on the basis of the score of 100/100/1/99 for SH-alrt/lbt/abayes/ufboot. Three of the six remaining ASVs were categorized in the clades representing genera in Family Suessiaceae, two ASVs were in the Yihiella clade (OTU.sym11 and OTU.sym13), and one was in the Ansanella clade (OTU.sym20). The remaining three ASVs were designated to the Suessiaceae family but were not classified at the genus level (OTU.sym3, OTU.sym9, and OTU.sym14).

Figure 3 Maximum likelihood phylogenetic tree based on V9-18S rRNA gene for Order Suessiales (A) and Family Symbiodiniaceae (B).

(A) Phylogeny of families in Order Suessiales. ASVs from this study (OTU.sym1–OTU.sym22) are shown in bold black font, and the branch support values represent the multi scores of SH-alrt/lbt/abayes/ufboot. Top: Motile stage of Symbiodinium natans and coccoid form of Symbiodiniaceae (source: LaJeunesse (2020)), Middle: Ventral view of Borghiella dodgei (source: (Pandeirada, Craveiro & Calado, 2013), Bottom: Ventral view of Polarella glacialis (source: Montresor et al., 2003). (B) Phylogeny of genera in Family Symbiodiniaceae. ASVs are shown in bold black, and red circles represent branch support scores >50 in SH-alrt. Phylogenetic reconstruction was performed in IQ-TREE and visualized with iToL (https://itol.embl.de/).

The Symbiodiniaceae family branch (Fig. 3B) comprised six clades, each representing one genus with strong-to-moderate support (see the scores in Table 3 and Fig. 3). This phylogenetic topology is concordant with the Symbiodiniaceae phylogeny reconstructed by Decelle et al. (2018). One ASV was allocated to each of clades Symbiodinium (A.sym21), Breviolum (B.sym18), Foraminifera Clade G (G2.sym4), and Halluxium (H.sym12). Eight ASVs were designated to Cladocopium (C.sym1, C.sym5, C.sym7, C.sym8, C.sym10, C.sym15, C.sym16, and C.sym17), and four ASVs allocated to Durusdinium (D1.sym2, D1.sym6, D1.sym19, and D1.sym22).

Symbiodiniaceae distribution and diversity

Venn diagrams show the overlap of the 16 ASVs belonging to Symbiodiniaceae according to location (Fig. 4A) and sample type (media and fractions) (Fig. 4B) (see also File S6). The presence/absence table shows the Symbiodiniaceae proportion per subclade by site–sample type–filter pore size combination (Table 4 and File S7). This table illustrates the common and unique subclades of Symbiodiniaceae. The unique subclades were the sequences distinctive of sampling location, medium, and fraction. Three subclades were most common (C.sym1, D1.sym2, and G2.sym4), and the remaining subclades were unique (Table 4). The unique subclades (<11.11% of subclade presence in all samples) showed site- or sample type-specificity. C.sym1 was the most common (77,78%) and was detected at more sites–media–fractions than D1.sym2 (44.44%) and G2.sym4 (33.33%). In term of medium, the sediment samples yielded more Symbiodiniaceae subclades than seawater (12 vs. 7 subclades), with nine unique ASVs found in the sediment medium.

Figure 4 Venn diagram of Symbiodiniaceae subclades around Lombok by: (A) coastal area and (B) method (sample type–filter pore size combination).

Sample labels: sea = seawater sample; sed = sediment sample; _0.4 and _12 indicate the pore size of the filter (in µm).

Table 4 List of Symbiodiniaceae proportion per subclade based on present/absent analysis by site–sample type–filter pore size combination.

Symbiodiniaceae types considered as “common” were ≥33.33% in the presence of all sample combinations, and unique were <33.33% in the presence of all sample combinations. Sample label: ESea0.4 indicate site–sample type–filter pore size combination of East Lombok_Sea Water_0.4–12 µm; ESea12: East Lombok_Sea Water_>12 µm; ESed0.4: East Lombok_Sediment_0.4–12 µm; ESed12: East Lombok_Sediment_>12 µm; NSea0.4: North Lombok_Sea Water_0.4–12 µm; NSea12: North Lombok_ Sea Water_>12 µm; NSed0.4: North Lombok_Sediment_0.4–12 µm; NSed12: North Lombok_Sediment_>12 µm; WSed0.4: West Lombok_Sediment_0.4–12 µm.

Intersection inter site-sample type-fraction	Proportion per subclade (%)	Total	Subclades	Type	
ESea0.4/Esed0.4/Esed12/NSea0.4/NSed0.4/NSed12/WSed0.4	77.78	1	C.sym1	Common	
ESea12/ESed12/NSea0.4/NSed0.4	44.44	1	D1.sym2	Common	
ESea12/Esed0.4/NSed0.4	33.33	1	G2.sym4	Common	
ESea0.4	11.11	1	C.sym16	Unique	
ESea12	11.11	1	D1.sym6	Unique	
ESed0.4	11.11	3	C.sym7	Unique	
	11.11		C.sym17	Unique	
	11.11		C.sym10	Unique	
ESed12	11.11	1	C.sym8	Unique	
NSea0.4	11.11	1	D1.sym19	Unique	
NSea12	11.11	1	H.sym12	Unique	
NSed0.4	11.11	3	D1.sym22	Unique	
	11.11		C.sym15	Unique	
	11.11		B.sym18	Unique	
WSed0.4	11.11	2	A.sym21	Unique	
	11.11		C.sym5	Unique	

On the basis of Symbiodiniaceae relative abundance, genus Cladocopium was the most dominant (Fig. 5). In general, the Symbiodiniaceae communities of Lombok were characterized with low alpha diversity and high beta diversity (Fig. 6). However, comparison of Symbiodiniaceae abundances, observed features, and diversity does not show significant difference between locations, media, and fractions (see File S8).

Figure 5 Composition of the relative abundance of Symbiodiniaceae communities across different sites, sample types, and fractions.

Relative abundance based on the total presence of ASV frequencies. Bar graphs represent the total percent abundance of Symbiodiniaceae detected from all samples. Sample labels: sea = seawater sample; sed = sediment sample; 0.4–12 µm and >12 indicate the pore size of the filter (in µm) sample.

Figure 6 Total diversity of Symbiodiniaceae in coral reefs waters around Lombok Island: (A) alpha diversity and (B) beta diversity.

Alpha diversity is indicated by Shannon index and beta diversity is represented by Bray-Curtis (BC) dissimilarity. Boxplots display the median as the midline, and the upper and lower quartiles as the top and bottom lines of the boxes, respectively. Cross symbols indicate the mean, and circles denote the outliers.

Discussion

The results illustrate the potential of eDNA to detect Symbiodiniaceae. The eDNA of Symbiodiniaceae can be obtained from different sources including free-living Symbiodiniaceae (Hirose et al., 2008; Littman, Van Oppen & Willis, 2008) and Symbiodiniaceae living in symbioses with various host organisms (Freudenthal, 1962; Loh, Cowlishaw & Wilson, 2006; Barneah et al., 2007; LaJeunesse et al., 2010a; LaJeunesse et al., 2010b; LaJeunesse et al., 2018; Pochon & Gates, 2010; DeBoer et al., 2012; Pochon, Putnam & Gates, 2014; Ramsby et al., 2017). Additionally, these eDNA sources could come from within and outside the sample site (Goldberg et al., 2016). Symbiodiniaceae DNA could be obtained from prey organism feces and through the shedding of host cells in the water and sediment (Rees et al., 2014; Grupstra et al., 2021).

The SSU 18S rRNA gene primer set has long been used in the biomolecular studies of Symbiodiniaceae (Rowan & Powers, 1991; Loh, Cowlishaw & Wilson, 2006). Hypervariable regions V4 and V9 isolated and then amplified by the SSU 18S rRNA gene universal primer were successful in detecting and identifying Symbiodiniaceae from water samples (Stoeck et al., 2010). This study used the same V9-SSU 18S rRNA gene primer set for oceanic planktonic Symbiodiniaceae by the Ocean TARA Expedition. The substitutions in the hypervariable terminal loop region amplified by this primer allowed us to distinguish Symbiodiniaceae genera and subclades (Decelle et al., 2018). Other primers such as ITS, LSU 28S, and chloroplast primers can be used to provide high taxonomic resolution for Symbiodiniaceae (Venera-Ponton et al., 2010; Takabayashi et al., 2012; Arif et al., 2014). Nevertheless, this study succeeded in detecting and identifying Symbiodiniaceae at the genus level.

The use of universal eukaryote primers with eDNA samples can reveal information on the rich diversity of marine life and compensate for the high cost of next-generation sequencing (Smart et al., 2016; Bálint et al., 2018). Universal primers allow us to broadly look at the system and complete more than a single study using the same data (Madduppa et al., 2021). The lack of field blanks (non-reef sampling areas) and filter blanks (distilled water or sterile seawater samples), might influence our study results. Lack of control/blanks can lead to contamination of the eDNA source, or false-positive data. However, the comparitive analyses across the given samples allowed the evaluation of the possibility of exogenous and local eDNA sources. Moreover, the presence of contaminant DNAs was likely suppressed by rinsing the instruments (e.g., bottle samples and filtering tools) with bleach to make them as sterile as possible.

To the authors’ knowledge, this work is the first study of Symbiodiniaceae using eDNA in Indonesia and Southeast Asia. Symbiodiniaceae in the Southeast Asia region have been identified from scleractinian stony corals, sea slugs, giant clams, and other bivalves, sea anemones, sponges, zoantharians, antipatharian black corals, and Heliopora blue corals. At least seven Symbiodiniaceae genera have been discovered in Southeast Asia (Table 5). Various primers, such as nuclear primers, mitochondrial organelle primers, and chloroplast primers, and a range of molecular techniques such as single stranded conformational polymorphism, restriction fragment length polymorphism, and denaturing gradient gel electrophoresis, have been used in the identification and characterization of the genetic diversity of Symbiodiniaceae in the region but did not detect as many genera as the present study did (see Table 5). No report was found about the genus Effrenium and Clade I in Southeast Asia. However, clade E (AF238261.1) in our phylogeny (Fig. 3) was assigned to clade D1 by Kimes et al. (2013). E. voratum is the only species from Effrenium that was previously described and is only found in temperate waters (Jeong et al., 2014). LaJeunesse, Parkinson & Trench (2012) predicted that the Southeast Asia region might have a higher diversity of Symbiodiniaceae species than other regions in the world. Previous and current findings supports this prediction (Loh, Cowlishaw & Wilson, 2006; Bo et al., 2011; DeBoer et al., 2012; Purnomo, 2014). Therefore, other under-sampled coral reef areas in Indonesia should be further explored.

Table 5 Comparison of Symbiodiniaceae studies in Indonesia and Southeast Asia region.

Geographic scope	Sample type(s)	Identification Method(s)	Genera	Reference(s)	
Indonesia:					
Sulawesi	Sea slugs (Pteraeolidia ianthina)	SSU 18S rRNA, LSU 28S rRNA, Single StrandedConformational Polymorphism (SSCP)	Cladocopium, Durusdinium	Loh, Cowlishaw & Wilson (2006)	
West Sumatra	Anthipatharian black corals (Cirrhipathes sp.)	ITS2 rRNA, LSU 28S rRNA, denaturing gradient gel electrophoresis (DGGE), restriction fragment length polymorphisms (RFLPs)	Gerakladium	Bo et al. (2011)	
Papua & West Papua	Giant clams (Tridacna spp.)	ITS2 rRNA, DGGE	Symbiodinium, Breviolum, Cladocopium	DeBoer et al. (2012)	
Central Java	Scleractinian corals, sea anemones, Tridacna sp.	SSU 18S rRNA, RFLPs	Symbiodinium, Breviolum, Cladocopium, Durusdinium	Purnomo (2014)	
West Nusa Tenggara	Seawater, sediment	V9-SSU 18S rRNA	Symbiodinium, Breviolum, Cladocopium, Durusdinium, Gerakladium, Halluxium	This Study	
Southeast Asia:					
Palau	Sponges (porifera), giant clams (Tridacna spp.), other bivalves (cardiids), foraminifera (Amphisorus hemprichii)	SSU 18S rRNA, RFLPs	Symbiodinium, Cladocopium, Durusdinium	Carlos et al. (1999)	
	Scleractinian corals	ITS1 rRNA, SSCP	Cladocopium, Durusdinium	Fabricius et al. (2004)	
	Scleractinian corals (Porites cylindrica)	ITS2 rRNA, psbAncr	Cladocopium, Durusdinium	Kurihara et al. (2021)	
Singapore	Sea slugs (Pteraeolidia ianthina)	SSU 18S rRNA, LSU 28S rRNA, SSCP	Cladocopium, Durusdinium	Loh, Cowlishaw & Wilson (2006)	
	Zoantharians	mt 16S rRNA, mt COI, ITS rRNA	Cladocopium, Durusdinium	Reimer & Todd (2009)	
	Scleractinian corals (Porites lutea)	ITS2 rRNA	Symbiodinium, Cladocopium, Durusdinium	Tan et al. (2020)	
Malaysia	Scleractinian corals (Porites lutea)	ITS2 rRNA	Symbiodinium, Cladocopium, Durusdinium	Tan et al. (2020)	
Thailand	Scleractinian corals, Corallimorpharia sp., sea anemones (Actiniidae & Stichodactyliidae), soft coral (Alcyonidae & Nephtheidae), gorgonian (Gorgonia sp.), giant clams (Tridacna crocea), Zoantharia (Palythoa sp.)	ITS1 rRNA, ITS2 rRNA, DGGE, microsatellite,	Symbiodinium, Cladocopium, Durusdinium, Fugacium, Gerakladium	LaJeunesse et al. (2010a) and LaJeunesse et al. (2010b)	
Philippines	Giant clams (Hippopus hippopus & Tridacna crocea)	SSU 18S rRNA, RFLPs	Symbiodinium	Carlos et al. (1999)	
	Heliopora blue corals (Heliopora coerulea)	SSU 18S rRNA, RFLPs	Cladocopium	Taguba, Sotto & Geraldino (2016)	
	Scleractinian corals (Acropora spp.)	ITS2 rRNA, DGGE	Cladocopium	Ravelo & Conaco (2018)	
	Scleractinian corals	ITS2 rRNA, DGGE	Cladocopium, Durusdinium	Da-Anoy, Cabaitan & Conaco (2019)	
South China Sea	Scleractinian corals	LSU 28S rRNA	Cladocopium, Durusdinium	Tong et al. (2018)	
Timor-Leste	Scleractinian corals	mt cob, psbAncr	Cladocopium, Durusdinium	Brian, Davy & Wilkinson (2019)	

The detected Symbiodiniaceae in the study sites are probably coral endosymbionts. Some species of Symbiodinium, Breviolum, Cladocopium, and Durusdinium are the main coral endosymbiont genera, and species of Fugacium and Gerakladium are rare endosymbionts in corals (LaJeunesse et al., 2010a; LaJeunesse et al., 2010b; Rouzé et al., 2017). The main coral endosymbionts, especially in Indo-Pacific, are species of Cladocopium and Durusdinium; meanwhile members Symbiodinium and Brevolium are common in corals in the Caribbean (Baker, 2003; LaJeunesse et al., 2004; LaJeunesse et al., 2010a; LaJeunesse et al., 2010b; LaJeunesse, 2005; Stat & Gates, 2011). Many members of Cladocopium (e.g., ITS2 subclade C1) generally have high rates of carbon fixation, provide a high fitness benefit, translocate high amounts of carbon to host corals, and positively impact host coral growth rates. By contrast, some species of Durusdinium tend to be opportunistic, even though they can help corals to survive or quickly recover from bleaching when sea surface temperatures rise (Stat, Morris & Gates, 2008; Stat & Gates, 2011; Lesser, Stat & Gates, 2013; Bay et al., 2016).

This study detected the three most common subclades, namely C.sym1, D1.sym2, and G2.sym4. These subclades may represent the most common species or types of Symbiodiniaceae. BLAST results showed that C.Sym1 was similar to C. goreaui (99.24%), formerly clade C type C1, which is a generalist Symbiodiniaceae found in many coral hosts in the Great Barrier Reef (LaJeunesse, 2005; Bongaerts et al., 2015). The sequence of D1.sym2 detected by BLAST has 100% sequence similarity with the molecular marker of D. trenchii, a Symbiodiniaceae species that increases the tolerance of corals to bleaching stress (Stat & Gates, 2011). Previous studies have suggested the importance of a minimum density of D. trenchii as a minority component alongside a dominant endosymbiont from the genus Cladocopium in the Symbiodiniaceae community within a coral colony (Bay et al., 2016). However, Swain et al. (2017) found that each genus of Symbiodiniaceae has the potential for heat-resistant species or variants. For example, C. thermophilum is a thermotolerant variant of Cladocopium type C3 (Hume et al., 2015).

This study fully resolved the ASV of subclade G2.sym4 within the Foraminifera Clade G (formerly clade G type G2). This genus can be isolated from the foraminifera, particularly in Subfamily Soritinae (Pochon et al., 2007). Bo et al. (2011) also isolated a subclade close to type G2 from Indonesian octocorals. Foraminifera Clade G is a common endosymbiotic Symbiodiniaceae in sponges, such as bio-eroding sponge (Cliona orientalis) in Australia (Schönberg & Loh, 2005; Ramsby et al., 2017). However, G2.sym4 appears to be a common type and is also found in the sediment samples. Therefore, this subclade may be an endosymbiont of benthic foraminifera. Foraminifera communities around Lombok are diverse, widely distributed, and present in the seabed in shallow coastal waters around the island (Auliaherliaty, Dewi & Priohandono, 2004; Natsir, 2009; Natsir, 2010; Dewi et al., 2012). However, no studies of foraminifera endosymbiotic Symbiodiniaceae in Indonesia have been published.

The detected Halluxium in this study is the first record in the Southeast Asia region. To date, Halluxium has only been found in Guam, Heron Island (Great Barrier Reef, Australia), and the Caribbean (Pochon, LaJeunesse & Pawlowski, 2004; Pochon et al., 2007; Nitschke et al., 2020). This genus and Clade I are generally foraminifera-specific endosymbionts. Meanwhile, Breviolum or Effrenium species living as foraminifera endosymbionts have never been reported (Pochon & Pawlowski, 2006; Pochon & Gates, 2010).

The richer Symbiodiniaceae subclades in sediment than in seawater indicate the potential occurrence of benthic Symbiodiniaceae. These Symbiodiniaceae can have important implications for the coral reef ecosystems of Lombok. The benthic sediment can be a source of free-living Symbiodiniaceae that live outside the host (Hirose et al., 2008; Littman, Van Oppen & Willis, 2008; Fujise et al., 2021). Some of these can (re-) establish stable host–algal mutualisms (transient free-living), and others are true free-living, such as E. voratum (Yamashita & Koike, 2013; Jeong et al., 2014). Some transient free-living Symbiodiniaceae can come from expelled coral endosymbionts. Corals regularly expel some of their endosymbionts into the seawater column (Fujise et al., 2014), most of which are deposited in sediments. The other source of transient free-living Symbiodiniaceae is reef fishes. Corallivorous, detritivorous, and herbivorous fishes can contribute to the release and distribution of transient free-living Symbiodiniaceae in their habitat through their feces (Castro-Sanguino & Sánchez, 2012; Grupstra et al., 2021). The availability of such Symbiodiniaceae in the environment is essential. During larval stage and/or recruitment time, most corals horizontally obtain transient free-living Symbiodiniaceae from the nearby environment (Coffroth et al., 2006; Fujise et al., 2021). The presence of such Symbiodiniaceae can also influence juvenile coral survival (Suzuki et al., 2013).

This study found that 13 of the 16 subclades were distinctive of different sampling locations. These subclades may represent the species or types of Symbiodiniaceae originating from local sources. Environmental genetic materials are prone to degradation (Barnes & Turner, 2016), so they tend to accumulate around the source. Therefore, eDNA is representative of the local biotic genetic material. Shinzato et al. (2018) showed the feasibility of studying nearby coral species and their symbiotic algae detection using eDNA; therefore, it might also be used to monitor coral ecosystem health. However, such data must be carefully interpreted because of some issues regarding the possible sources of eDNA from outside the sample site due to biological factors and human activities (Goldberg et al., 2016).

The eDNA method also has some limitations, such as the dependence on the presence and concentration of eDNA in the water sample, capture efficacy, extraction efficacy, sample interference (e.g., inhibition), and assay sensitivity (see Goldberg et al., 2016). Seawater eDNA samples can degrade beyond the detection threshold within 1 day to weeks (Dejean et al., 2011; Thomsen et al., 2012). Water quality conditions, such as high temperatures, neutral pH, and moderately high UV-B, tend to increase the eDNA degradation rate (Strickler, Fremier & Goldberg, 2014). However, the degradation rate of eDNA in aquatic environments is different from that in sediments. The nature and proportion of minerals, organic substances, and charged particles adsorbing eDNA fragments influence the rate of eDNA degradation in sediments and protect them from further destruction. A previous study showed that the degradation rate of eDNA in sediment is about 57 times slower than that in seawater (Torti, Lever & Jørgensen, 2015; Turner, Uy & Everhart, 2015; Sakata et al., 2020). Limited information is available regarding the factors that influence the rate of symbiont DNA shed by coral reef taxa and maintained in the water column over spatial scales.

Conclusions

This study demonstrates that eDNA surveys can describe the potential diversity of Symbiodiniaceae in the reefs around Lombok. Six genera (or genera-equivalent clades) of Symbiodiniaceae were identified. eDNA survey has higher sensitivity than traditional methods and thus offer a rapid proxy for evaluating Symbiodiniaceae communities across different coral reefs. This approach can also be used to enhance the understanding of the diversity and relative ecological dominance of certain Symbiodiniaceae members. Moreover, the presence of distinctive Symbiodiniaceae individuals in different locations support the potential application of eDNA for monitoring the local and regional stability of coral–algal mutualisms. Further confirmation through isolation from a variety of sources (including possible hosts) and microscopic observations is warranted to strengthen the evidence for local eDNA sources.

Supplemental Information

Figure S1 Count of sequences (raw data) per sample

Click here for additional data file.

File S1 OTU sequence data

Click here for additional data file.

File S2 Taxonomy of Eukaryote at 97 % and 99 % similarity and putative Symbiodiniaceae sequences

Click here for additional data file.

File S3 BLAST results from putative Symbiodiniaceae OTUs

Click here for additional data file.

File S4 Final compiling reference sequence database for phylogenetic reconstruction

Click here for additional data file.

File S5 Relative abundance data of putative Symbiodiniaceae

Click here for additional data file.

File S6 Detailed explanation of unique elements at Venn diagram of Symbiodiniaceae subclades around Lombok

Click here for additional data file.

File S7 Prensence of Symbiodiniaceae based on site-medium-fraction sample combination

Click here for additional data file.

File S8 Raw data for diversity

Click here for additional data file.

This manuscript is dedicated to the memory of our dear colleague, friend, and dive buddy Dr. Hawis Madduppa. We will always remember your infectious smile, curiosity for the marine environment, and confident leadership. The authors thank all staff and research colleagues who are members of the Center for Collaborative Research ANBIOCORE (Animal Biotechnology and Coral Reef Fisheries), Bogor Agricultural University, the University of Rhode Island research team for their assistance during the research, and the Laboratory of Marine Biodiversity and Biosystematics. The authors are also grateful to Prof. Dr. Ir. Sudarsono, M.Sc. for technical preparation to the manuscript draft.

Additional Information and Declarations

Competing Interests

Author Contributions

Field Study Permissions

Data Availability

The authors declare there are no competing interests.

Arief Pratomo conceived and designed the experiments, performed the experiments, analyzed the data, prepared figures and/or tables, authored or reviewed drafts of the article, and approved the final draft.

Dietriech G. Bengen analyzed the data, authored or reviewed drafts of the article, and approved the final draft.

Neviaty P. Zamani analyzed the data, authored or reviewed drafts of the article, and approved the final draft.

Christopher Lane conceived and designed the experiments, performed the experiments, authored or reviewed drafts of the article, and approved the final draft.

Austin T. Humphries conceived and designed the experiments, performed the experiments, authored or reviewed drafts of the article, and approved the final draft.

Erin Borbee conceived and designed the experiments, performed the experiments, analyzed the data, authored or reviewed drafts of the article, and approved the final draft.

Beginer Subhan performed the experiments, authored or reviewed drafts of the article, and approved the final draft.

Hawis Madduppa conceived and designed the experiments, performed the experiments, analyzed the data, prepared figures and/or tables, authored or reviewed drafts of the article, and approved the final draft.

The following information was supplied relating to field study approvals (i.e., approving body and any reference numbers):

eDNA seawater sampling in this study was permitted within the framework of the United States Agency for International Development—Sustainable Higher Education Research Alliances (USAID-SHERA) program through the Centre for Collaborative Research Animal Biotechnology and Coral Reef Fisheries (CCR ANBIOCORE) of IPB University, award no. AID-497-A-16-00004. The field research permit was issued by IPB University Rector (Surat Tugas no. 403/IT3/KP/2019). Permits for this research were issued by the Indonesian Ministry of Research and Technology to EB (130/E5/E5.4/SIP/2019), CL and AH (455/SIP/FRP/E5/Dit.KI/XII/2017).

The following information was supplied regarding data availability:

The raw sequence data is available at GenBank: PRJNA768103.

The outative Symbiodiniaceae OTU sequence data is available at GenBank: SRP339775.

The code for this research analyses is available at GitHub: https://github.com/arief2021/Symbio_Qiime2.git.

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
