# Peer review of "Diversity and distribution of Symbiodiniaceae detected on coral reefs of Lombok, Indonesia using environmental DNA metabarcoding"

_PeerJ, doi:10.7717/peerj.14006_

## Round 0.1 · original submission · Major Revisions

Dear authors,

I have received three independent reviews of your study. While all reviewers clearly recognized the novelty and potential of your work, they have collectively raised a number of major issues that will need to be addressed in your revised manuscript. In particular, all reviewers have highlighted that the current manuscript will require some important reformulation in hypotheses, analyses, presentation and conclusions, which are currently not supported by the data.

All reviewers have also pointed to the very limited resolution afforded by the 18S marker and the inappropriate use of 'generalists vs specialists' Symbiodiniaceae species in this context. Also, clustering ASVs into OTUs should be avoided.

Overall, the reviewers have provided you with exceptionally good suggestions on how to improve the manuscript, and I really hope that you will tackle this revision in the same constructive spirit as the reviewers did. I will be looking forward to receiving your revised manuscript along with a point-by-point response to their comments.

With warm regards,
Xavier

Reviewer 1 ·

Excellent Review

This review has been rated excellent by staff (in the top 15% of reviews)
EDITOR COMMENT
I commend this reviewer for their time and efforts in providing excellent review comments that will undoubtedly help authors improve their work and produce a high-quality paper that will represent a valuable contribution to the field of coral reef Science. Thank you! Xavier

Basic reporting

1. This study reveals the diversity of Symbiodiniaceae in water and sediment samples from around Lombok, Indonesia, using high throughout sequencing of the V9 hypervariable region of 18S rRNA gene. Twenty-two OTUs of Symbiodiniaceae were identified and classified using various approaches. Some of these OTUs were reported to be common across sites and some that were found only in some samples.

2. Overall, the study presents interesting data that contributes to our current knowledge of Symbiodiniaceae diversity in the Indo-Pacific. However, data treatment and presentation need to be improved.

3. The paper is concise and is written clearly. There are just a few sections that should be rephrased to clarify ideas and other sections that should be reorganized or expanded to facilitate comprehension (details are provided in the section for additional comments).

4. Minor grammatical/typographical errors throughout the manuscript should also be double checked.

Experimental design

1. The rationale for looking at Symbiodiniaceae in water and sediment, as opposed to host organisms (as most other studies have done), should be more clearly articulated.

2. The sampling design and methods for sample processing were presented in detail. However, it was not clear from methods how the data from the different stations was treated in the downstream analyses.

3. Reasons for using 2 filter sizes and for choosing 18S rRNA V9 primers for sequencing should be clearly explained.

4. Other key details are missing from methods.

Validity of the findings

1. Information on replication and treatment of data from stations within each sampling site should be clearly stated.

2. It is suggested that authors analyze their data at the ASV level. By clustering ASVs into OTUs, information on finer sequence variability is lost, and this information could actually be quite interesting.

3. The paper focused on the few OTUs that were affiliated with Symbiodiniaceae but did not present the other data derived from their eDNA surveys, which would have been interesting to see, as well.

4. Phylogenetic trees shown in Figures 2 and 3 are somewhat redundant.

5. The Venn diagram analysis does not present the data sufficiently. Statistical analyses to compare Symbiodiniaceae sequences detected across different sites (East, West, North Lombok) or sample types (water, sediment) would greatly strengthen this paper. It would also be interesting to see the relative abundance of detected ASVs across sites/samples.

6. The use of the terms generalist and specialist in the context of describing the presence or absence of OTUs across sites and sample types is inaccurate. It would be better to use the terms common or unique.

7. Raw sequence data should be deposited in NCBI or other public repository and accession codes indicated here. Processed data (e.g. ASV and OTU sequences, relative abundance matrices, etc.) should be publicly accessible, as well, if not yet included in supplemental materials.

Additional comments

Abstract

Line 22-23: Omit “due to technical difficulties.” In fact, there have been many studies that looked at Symbiodiniceae function and distribution globally and in the region, though these are mostly focused on populations within host organisms. Authors should emphasize how their study differs from these and why it is important to survey symbionts in the water and sediment.

Line 30: It is more appropriate to say “rRNA gene” rather than “rDNA”. This should be corrected here and throughout the manuscript.

Line 36: Criteria for classification in generalist vs specialist is not clear


Introduction

Line 63: Omit “The”

Line 69: “This family” > “Symbiodiniaceae”

Line 83-86: Specify what types of studies these were or mention a little bit about their findings in order to highlight research gaps addressed in this paper. Note that next-generation sequencing approaches have also been used successfully in community profiling of Symbiondiniaceae, typically from animal hosts, globally and in the Indo-Pacific area.

Line 93-97: The rationale for the eDNA approach for Symbiodiniaceae assessment from water and sediments needs to be better articulated. For example, would diversity of Symbiodiniaceae as determined from the environment likely reflect communities present in coral on the reef? This would allow authors to make a more direct link to coral reef conservation and management applications.


Methods

Line 101-103: How many samples of water/sediment were collected from each station (replicates)?

Line 105: How were sediment-associated cells extracted and were there additional steps done before filtration of the sediment/water samples through the filters?

Line 108: What was the rationale for using these 2 different filter sizes?

Line 108: “places” > “placed”

Line 118: Omit “universal”

Line 120: Why was the 18S rRNA V9 region selected for this analysis? For the purpose of resolving Symbiodiniaceae diversity, ITS2 is usually the preferred region.

Line 125-126: Indicate final concentrations of reagents in the PCR reaction. Indicate source companies for all reagents used.

Line 127: “0.5” > “0.5X”

Line 147: Do the authors mean “< 20” instead of “> 20”?

Line 158: It's not clear how this was done. Did they classify first at 97% and then more specifically at 99% or did they use the classification at 99% but used the 97% classification for OTUs that could not be classified at the more stringent level? Please clarify.

Line 187: “resulting” > “results”

Line 189-193: What were the criteria used for assigning OTUs as generalist or specialist? Some sort of statistical analyses to compare Symbiodiniaceae detected across different sites (East, West, North Lombok) or sample types (water, sediment) would greatly strengthen this paper.


Results

Line 207: What were the affiliations of the other ASVs? What proportion of the data was classified as Symbiondiniceae? Given what we know about the diversity of Symbiodiniaceae and the power of next-generation sequencing, it would be good to analyze the data at ASV level to truly showcase diversity in the reefs that were sampled.

Line 210-211: The sentence “During the processing, it was also found that Symbiodinium sp. clade E (AF238261.1) had been revised and assigned to clade D1 by Decelle et al. (2018)” does not fit into the results and should perhaps be mentioned in Discussion instead.

Line 220: How do you explain the two separate groups of Durusdinium in the tree with one group containing only clade D sequences and another clustering with sequences from clades G and F?

Line 223-228: Clarify phrasing of these sentences. Maybe something like "A single OTU was nested within each of clades Symbiodinium/clade A (A.Sym21), Breviolum/clade B (B.Sym18), Gerakladium/clade G2, and Halluxium/clade H (H.Sym12). Eight OTUs nested within Cladocopium/clade C (C.Sym1, C.Sym5, C.Sym7, C.Sym8, C.Sym10, C.Sym15, C.Sym16, and C.Sym17) and 4 OTUs within Durusdinium/clade D1 (D1.Sym2, D1.Sym6, D1.Sym19, and D1.Sym22).”

Line 232: Venn diagrams do not accurately reflect what is mentioned in this section as they show just 1 OTU common to all sites and/or sample type. It is suggested that a figure be included that shows relative abundance and diversity of OTUs across sites and sample types. Consider conducting statistical analyses to compare Symbiodiniaceae OTUs across sites or samples.

Line 235-336: It is suggested that instead of using the terms generalist and specialist, the authors use “common” or “unique” to refer to distribution of detected OTUs across sites and sample types.


Discussion

Lines 249-252: This paragraph initially began by discussing the eDNA of Symbiodiniaceae, but by line 249 it shifted to discussing the sources of eDNA in general. It would also be best to contain the discussion of eDNA in one place to facilitate cohesiveness and comprehensibility.

Line 252: “The SSU 18S…” should be in a separate paragraph

Line 258: hyper variable > “hypervariable”

Line 270: Should note that most other studies focused on coral-associated symbionts, thus do not detect free-living clades or clades associated with less-studied hosts, such as foraminiferans. This could be highlighted here because this is one advantage provided by the eDNA approach in water and sediment samples.

Line 275-276: Provide additional details to compare these other studies in Table 5 to the current study. What samples were analyzed, what primers/methods were used, what clades were detected?

Line 280: “this study findings” > “the findings of this study”

Line 288: Not clear what is meant by a “balanced” symbiosis.

Line 290-291: It is suggested to use the term “common” instead of generalist and “site or sample-specific” instead of specialist in this context (although some groups, such as members of Cladocopium, have been detected in many regions and hosts, suggesting that they are indeed generalists).

Lines 303-308: The genus Halluxium, which was found by the study to be a specialist, was brought up during a discussion of generalist OTUs, making the discussion difficult to comprehend. Additionally, the author states that the detection of Halluxium in this study is interesting without clearly explaining why it is so.

Line 320-322: This is an interesting finding that was not mentioned in results. It would have been great to have a figure showing relative abundance and diversity of OTUs across sites and sample types.

Line 335: Limitations of the method should also be explicitly mentioned. For example, certain clades may not have been detected in certain samples because they could be in very low abundance. It is also possible that some cells are more susceptible to environmental degradation either due to intrinsic qualities or perhaps environmental conditions.

Line 337: How does the symbiont population detected by eDNA methods from water and sediment reflect symbiont populations in the corals on the reef? This connection needs to be made clear in order to be able to use eDNA detection was a means of monitoring coral ecosystem health, as mentioned in Line 339-340.


Conclusions

Line 347: Again, use of generalist and specialist in this context is inaccurate.


Data availability

Raw sequence data should be deposited in NCBI or other public repository and accession codes indicated here. Processed data (e.g. ASV and OTU sequences, relative abundance matrices, etc) should be publicly accessible, as well, if not yet included in supplemental materials.


Figures and tables

Figure 1: Check labels on panels B and C

Figure 2: “rDNA” > “rRNA”; spell out ML; mention in figure legend what the pictures on the tree represent and where they were obtained.

Figure 2 and 3: These are essentially the same data presented 2 different ways. It is suggested that just one of these figures be retained in the main text and the other moved to supplement.

Figure 4: From these figures, it looks like there is only 1 true “generalist” OTU that is found in all sites/sample types. A relative abundance plot of OTUs/ASVs in each sample and/or sample type would perhaps be more informative.

Table 3: First column header should be OTUs. Can add a line to separate the top header from the bottom headers in the other columns.

Table 4: From this table, it is not clear what criteria was used to designate OTUs as generalist/specialist. Only C.sym1 is truly common to most sites and samples, while D1.sym2 and G2.sym4 were not found in West Lombok samples. Was this data derived by combining the data from all stations per site? It is still unclear why the 0.4 and 0.12um filtered samples are presented separately.

Table 5: Could also indicate here what method as used for Symbiodiniaceae identification.

Reviewer 2 ·

Excellent Review

This review has been rated excellent by staff (in the top 15% of reviews)
EDITOR COMMENT
I commend this reviewer for their time and efforts in providing excellent review comments that will undoubtedly help authors improve their work and produce a high-quality paper that will represent a valuable contribution to the field of coral reef Science. Thank you! Xavier

Basic reporting

The English of this paper is generally fine. References are partly sufficient, although some areas need better or more relevant references. The structure of the paper is also standard IMRaD format, and fine.

Experimental design

The paper is original, and falls within the scope of PeerJ. The research question is defined well, although as you can see in my additional comments, will need to be completely revamped. The investigation, will welcome as there are few data from this part of the world, is not rigorous enough for the questions the authors set out to answer. Materials and methods are also lacking some details (I detail this in the additional comments section).

Validity of the findings

Conclusions are not supported by their data or analyses. See additional comments for details.

Additional comments

Overview:
The paper examines Symbiodiniaceae diversity from water samples from coral reef environments from Indonesia. Data on these symbionts from this part of the world are scarce, and thus the paper is a welcome addition to the literature and should help fill some gaps in our knowledge. Based on these reasons alone, I really hope this paper is accepted at some point in the future.
That said, there are several major concerns with this work as currently presented that preclude acceptance, and these concerns must be addressed to even consider the work as acceptable. I hope my strict comments do not discourage the authors but instead motivate them to improve this work. On the bright side, I am not suggesting generation of new data, but a reformulation in their hypotheses, analyses, and presentation.

Major concerns:
1. My largest concern is the hypotheses and conclusions. The authors state that they found both generalist and specialist Symbiodiniaceae in their samples, based on short 18S eDNA sequences (~127 bp). This kind of conclusion cannot be made from such short sequences and the rough cutoff levels used in this study. 18S does not resolve to species-level for Symbiodiniaceae, and this has been shown many times. Even the faster-evolving ITS region does not do so for Cladocopium, and thus, for the authors to conclude that they have specialist and generalist lineages in their samples is incorrect. The authors can only discuss genera and subclade identities, and should rightly highlight their Halluxium findings (for example), and present their work instead as a rough scan of diversity of Symbiodiniaceae in the examined areas, and also discuss what they found from what method (water/sediment), and nothing more. Similarly, assuming that their sequences are from host species such as corals, or free-living, also cannot be done with the current rough resolution of results. You can of course discuss this, but making conclusions on these matters is not correct given the data available.
Thus, based on this issue alone, it is my opinion that large portions of the Introduction, M&M, Results, and Discussion all need to be redone. By doing so, it makes any of the smaller comments I could offer unimportant or pointless to currently address, as any new version will be drastically different from the current version. Thus, I have not included many such comments in this review, instead focusing on the major issues that need to be addressed.
2. Additionally, the Materials and Methods is lacking a lot of the standard eDNA paper information I would want if I was to try and replicate this study. For example, I would add at least the following information:
a. How and why were your sites chosen? What is your rationale behind your experimental design? How deep were sites where water was chosen? At what depths were sediments collected? Were there field and filter blanks?
b. How soon after collection were samples filtered?
c. Were sediment samples filtered as well?
d. How were samples transported back to the lab (e.g. in a cooler box?) and how long did this take?
e. You say V9 - how large/long is your targeted area (bp)?
f. Did you combine or separate sediment and water samples? Why? How? It has been shown the results from these different types of samples are often very different (DiBattista et al. 2018, etc.). Did you take this into account in your analyses at all? Ahh = I can see you did to some degree.
g. Given past reports on the 18S and even the ITS marker having low resolution for resolving Symbiodiniaceae species, how and why did you choose the 97% and 99% similarity levels? You should reference something here. I personally would also examine ALL unique genotypes, and worry if you used these rough cutoffs that you might have missed a lot of potential diversity.
3. The nomenclature used does not rely on LaJeunesse et al. (2018) and subsequent papers, as you often use “clades” when generic names exist. I would use the generic names when possible, just mention clades once and then be done with them.
4. Why was DNA only successfully extracted from 41 of 72 samples? This seems low and suggests problems in preservation or extraction, and thus such information in the M&M is really needed to help readers assess this.

Minor points:
1. Line 22: Symbiodiniaceae do not regulate coral health. They contribute to it only.
2. Line 22: What do you mean by “technical difficulties”?
3. Line 25: In and around the coral reefs? Why are cardinal directions capitalized?
4. Line 33 and many other locations: “analysis” is used, but I would use plural “analyses”. Many instances of this. E.g. line 185
5. Line 34 and others: Use genera, not clades.
6. Lines 34-39: Cannot support these conclusions.
7. Lines 41-42: I do not know how the current results can support conservation and management? Can you explain this concretely?
8. Lines 43-44: See comment directly above.
9. Subjects: this work is not Taxonomy.
10. Lines 63-66: foraminifera should be capitalized (I think), also line 304, 305, 315, 316, 319. As you have written these sentences, it is odd and not correct. “Corals” – what do you mean by corals, do you mean scleractinians? – are also cnidarians, so your second sentence in this paragraph is somewhat redundant.
11. Line 69-78: This paragraph is oddly worded. Why not just say the genera, with clades in parentheses, and leave it at that. Please restructure these sentences. You also should include the subsequent works describing all the other genera, and clade J too!
12. Line 87: “not easy” – isn’t this almost impossible? Seems a bit oddly worded.
13. Line 92: For your proposed work, there may also be disadvantages, especially from water samples being used for coral symbiont examinations. You might want to mention these issues and some papers, either here or in Discussion.
14. Lines 96-97: Not really sure you can do that from current data, overstating things a bit?
15. Materials and methods: Asides from the issues in the Major concerns, some more comments below.
16. Line 106: Rinsed how? Filtered immediately on-site?
17. Line 155: I would change the title to Identification perhaps, to me “Assigning taxonomy” seems oddly worded.
18. Lines 182-183: A matching 18S sequence does not necessarily mean the same species, see my major comments. Also lines 294-299.
19. Lines 210-211: Sorry, I am not sure what you mean here?
20. Section from line 230: Cannot do this, see major comments.
21. Line 234: LaJeunesse.
22. Line 247: symbioses. Also, your references here do not include so many various hosts, why not cast your reference net wider?
23. Lines 252-256: Identifying – yes, but only roughly.
24. Line 260: clades again, should be genera. Also line 271, 275,
25. Lines 259-261: Here you even state the problems with 18S, and mention only to genus level (line 262-263).
26. Lines 264-267: This sentence runs on and seems to combine a few ideas. I would split this and make this section easier to understand.
27. Lines 268-269: First for all of SE Asia? I would check this to be certain.
28. Line 274: Heliopora, in italics.
29. Line 285: This sentence is inexact. Yes, there are genera of Symbiodiniaceae that associate with scleractinian corals, but in the Atlantic, Breviolum is important too, and the word “important” seems loaded with uncertain meaning. It is hard to make general conclusions at the genus-level, as you do on lines 288-290, and I would really avoid this kind of discussion given the rough level of your molecular data. You even state so on lines 299-302.
30. Lines 320-333: This paragraph also needs more nuance. It is my understanding that many free-living Symbiodiniaceae are exactly that, free-living, and not a source of symbionts for corals. The coral symbionts (symbiotic types) may be temporarily in the environment after being expelled or in parrotfish faeces, but do not live there. Much of this misunderstanding may come from older studies lumping different species together based on 18S or ITS sequences. I think many of your sentences in here are not quite exact or qualified in their meaning, so please reconsider your writing here.
31. Line 344: Conclusions (plural)
32. References: Missing italics, not consistently formatted in areas.
33. I would combine Figures 2 and 3.
34. Give depths in Table 1.
35. Table 5 – Many corrections needed. Give generic names please of Symbiodiniaceae. Also, some information missing here. “C, D” needed for the Reimer and Todd 2009 reference. There are information on Cladocopium in Acropora from Palau in Kurihara et al. 2020. Many common names here do NOT need to be capitalized (Scleractinian, Cardiids), while other names should be italicized (Heliopora). Palythoa spp. are not family Zoanthidae, but family Sphenopidae, or order Zoantharia.

·

Basic reporting

The paper submitted by Pratomo and co-authors is the first project to my knowledge that uses eDNA to study the presence of symbiodiniacean dinoflagellates in environmental samples. The research was conducted in an area of the Indo-Pacific were very little information is available on the diversity and ecology of these dinoflagellates.

This project relied on the use of a variable region of the 18S rDNA subunit. There is little information on this particular maker relative to other markers such as LSU rDNA and ITS2 rDNA, but the investigators seem to have been able to identify the presence of certain genera represented in their samples.

I am weary of their use of the term OTU because this is likely to lead to further confusion, as none of these are relatable to any existing database on OTUs designated based on other genetic markers. I think that it is perhaps best to halt the paper’s narrative at finding representatives of different symbiodiniacean genera in different samples from different locations around Lombok Island (i.e. keep it simple).

Their use of the word specialist and generalist is incorrect. These are ecological terms that mean something very different. They should change this wording to reflect the possibility that some (genotypes) were more frequently found across their surveyed locations then others. I don’t think that they have evidence to show with confidence that some are geographically widespread while others are endemic given the current narrow scope of sampling. This paper is a proof of concept and should be highlighted as such. Making any sweeping ecological claims is too premature.

The authors seem to be unaware of the existence of an eleventh genus Miliolidium (formerly a clade D sub-clade) Pochon X, LaJeunesse TC (2021) Miliolidium n. gen, a new symbiodiniacean genus whose members associate with soritid foraminifera or are free-living. 68:e12856

Is it possible that some of their “clade D” sequences are of this genus?

I think the manuscript would benefit from additional editing and simplification. For example the third paragraph is a bit tedious to the reader. I would begin this paragraph with the last sentence: “Currently, there are eleven recognized….” And proceed from there. The reader does not need to know about the convoluted historical baggage of arcane pseudo-taxonomy in the field.

I also recommend that the authors spend a little more time carefully choosing the papers they reference. For example the sentence in the Introduction: These Symbiodiniaceae are also associated with the adaptability and resilience of coral to environmental change, in particular global climate change impacts, especially global warming (Berkelmans & Van Oppen, 2006; Baskett, Gaines & Nisbet,2009). This papers choosen to substantiate this statement are a peculiar choice. There are many others that are more appropriate.

Instead of Citing LaJeunesse, Parkinson & Trench, 2012, cite LaJeunesse et al. 2018 Current Biology. Moreover they should reference this paper when talking about the family’s phylogeny instead of or along with Decelle et al. 2018.

Experimental design

No issues per se.

Validity of the findings

See above

Additional comments

See above

---

## Round 0.2 · Major Revisions

Dear Dr. Madduppa,

I have received another round of reviews from three reviewers, and have decided on another Major Revision. This is because the revised manuscript still requires careful English editing as well as corrections from incorrect statements and missing references.

Please note that reviewer #3 has provided you with comments and edits on your track-change version of the manuscript (attached). However, you will have to also incorporate all comments from Reviewers #1 and #2.

Overall, the reviewers have provided you with excellent suggestions on how to improve the manuscript, and I'll be looking forward to receiving your revised manuscript along with a point-by-point response to their comments.

With warm regards,
Xavier

Reviewer 1 ·

Basic reporting

The revised manuscript will require careful English language editing to improve clarity. Examples of sections that need attention are listed below:

Line 75: Suggest revising to “… 15 distinct lineages, with Foraminifera Clade G … and Clade I potentially representing separate genera.”
Line 83: Suggest revising to “Some Symbiodiniaceae genera reported in these studies include Symbiodinium …”
Line 87: Suggest revising to “Symbiodiniaceae cannot be directly identified using conventional microscopy.”
Line 222-224: Description of statistical method is unclear
Line 243-245: not clear what is meant by neither included nor classified as Symbiodiniacera
Line 271-272: “sample type-specificity”
Line 336: “Members of Cladocopium (e.g., ITS2 subclade C1), in general, have higher rates of carbon fixation…”
Line 355: “first record”
Line 403-404: Sentence needs to be rephrased. Maybe the authors meant “previous studies” instead of the current study?

Experimental design

Authors have added sufficient detail to their methods and have explained their experimental design.

Validity of the findings

Results are described in detail and conclusions are supported. Additional statistical analyses have been conducted. Limitations of the analyses are explicitly stated. Underlying data are provided in supplements and in links to databases.

Reviewer 2 ·

Basic reporting

The work is improved, but yet still far from publishable. English still needs work in areas (easily addressed), but more importantly, there are still ambiguous and incorrect statements throughout the text, and a lack of some references. In my opinion the work still needs a lot of harder work from authors before being acceptable. I have added detailed comments in the additional comments section.

Experimental design

Authors must discuss what the lack of field and filter blanks could mean for the results of their experiment.

Validity of the findings

Authors must discuss what the lack of field and filter blanks could mean for the results of their experiment.

Additional comments

Major concerns:

There are still inaccuracies throughout the text; the authors have edited areas that were pointed out in the previous round, but not really made a concerted effort to improve the paper in some areas. I have tried to add as many comments as I can below, but I would really like to ask the authors to please work a bit harder on bringing their paper up to modern Symbiodiniaceae standards to make it a welcome addition to the literature. Please note I have not made comments on a lot of the small English edits that are needed, and tried to focus on the science and story of the paper.

Comments (all line numbers refer to the tracked changes MSWord file)
1. Line 23: delete “due to laborious and time-consuming works,”
2. Lines 25-26: This is still ambiguous and not very concrete. Suggest deleting it.
3. Line 35: plural “analyses”. Also line 38; check throughout paper. Line 634, 345
4. Line 90: what do you mean by “unique” – this is never defined and I am not sure what you mean here.
5. Line 111: Are there other groups of endosymbiotic dinoflagellates on coral reefs? Odd statement to make here.
6. Line 112: Here you say “coral” and all throughout this paragraph, and then after discuss other animals. Why not go from general (in symbioses with many animals) and then focus on scleractinians (please define “corals”).
7. As well, how do you know your eDNA work only focuses on corals/scleractinians? This seems like a mismatch, shouldn’t your eDNA be examining Symbiodiniaceae not of corals but of coral reefs and all its symbiotic hosts??
8. Line 115: not only coral here
9. Line 136 and this paragraph: Again, please check the clade J paper (Yorifuji et al. 2021), it is still not included despite previous mention. As well, I would reword lines 141-142 as “, including some clades that represent undescribed genera”.
10. Line 146-148: This is a fragment, not a sentence. Lines 150-152 are also awkward and need to be rewritten.
11. Line 153: Rewrite this first sentence of this paragraph.
12. Line 186: I would argue this is not just “coral” but the entire reef ecosystem’s Symbiodiniaceae.
13. Line 187-88: Still weak, please be more concrete here.
14. Line 191: “Site” should be plural.
15. Line 193: “island”
16. Line 204: “western” and “southern”
17. Line 212: this is still not clear. Do you mean 18 samples per day? 3 sites, 6 samples per site, per day?
18. Line 215: how much sediment was collected?
19. Sorry, still not clear on the methods. You say each sample was 4L of SW, and you collected 6 samples? So you collected 6 X 4L of water? Please explicitly explain your methods here to leave no ambiguity.
20. Line 234: Please rewrite this sentence on sediment samples.
21. Note: subtitles are inconsistently capitalized.
22. Lines 294-296: I disagree with the choice of cutoffs as explained by the authors. Just because this is good for other protists does not mean it is good for Symbiodiniaceae, particularly given the wealth of information available for Symbiodiniaceae. You can of course still use these cutoffs, but at the least, some discussion about the adequacy and effects on your results is needed in the Discussion in such a case.
23. Lines 378-380: Again, you need to include in your paper why only 41 of 72 samples were extracted.
24. Line 397 and other areas: The authors use of “it” is a bit odd and hard to follow. Strict editing needed please.
25. Line 525: You need to explain “unique” here. Do you mean “unique in this study to one site”?
26. Line 632: Here I would say “genus Cladocopium was most dominant”. Also, shouldn’t results all be in past tense?
27. Line 699: scleractinian stony corals
28. Lines 701-705: I think it would be good to include references for each method, or at least refer to Table 5 here.
29. Note there are lots of simple English mistakes in this section, including but not limited to spelling errors, subject-verb agreement. Thorough checking is needed.
30. Line 775: Please be very careful with generic-level generalizations. In particular, Cladocopium contains a lot of species and types, and it is easy to oversimplify. As well, please remember most of these genera have types in association with animals besides “corals” too. Line 777 too is not really accurate. You can qualify these statements by saying “some species of….” for example.
31. Line 782: I do not know what they authors mean here by generalist, and cannot agree with this hypothesis. 18S cannot resolve to species, and it could just be that these “subclades” represent one or many types that were found at your sites. How these are generalist is not well explained, and in my opinion not correct.
32. Lines 844-onwards paragraph: Again, some nuance needed here. Free-living types are not the same as types expelled from hosts and temporally found in sediment; this distinction does not seem to be made here and you could read this section as saying they are the same. If there are free-living species, how are they essential to corals establishing symbioses? Logically, this does not make sense, and this section needs to be rewritten.
33. Line 857: Specialist? What are these? Again, like generalist above, I cannot agree with this hypothesis. I think the authors are trying to distinguish between types found at one site versus those found at many sites; new and more accurate terminology is needed.
34. On a positive note, the figures are improved and nice, and the tables well done. Table 5 alone is a very nice summary for researchers.

·

Basic reporting

I have edited the abstract for clarity and accuracy. I recommend that these changes be incorporated through out the manuscript to improve the narrative concerning the significance of the work and its future application.

Experimental design

none

Validity of the findings

See revised edits to the abstract.

Additional comments

none

---

## Round 0.3 · Minor Revisions

Dear authors,

Two reviewers have provided additional minor comments. It was noted that the English still needs a round of strict editing - if you could please incorporate all reviewers comments and have the revised manuscript proofread to remove English grammar mistakes - this would be much appreciated.

I'll be looking forward to receiving your revised manuscript along with a point-by-point response to the reviewers' comments.

With warm regards,
Xavier

Reviewer 1 ·

Basic reporting

The authors appear to have addressed most reviewer comments and incorporated suggested revisions into the manuscript. They have updated their references and improved the figures. However, while most of the manuscript is easy to understand, there is still some room to improve clarity and conciseness throughout. Specific suggestions are listed under additional comments below.

Experimental design

Authors have added sufficient detail to their methods and have explained their experimental design.

Validity of the findings

The revised objectives and results make more sense and highlights the usefulness of the study. Results are well described and conclusions are supported. Limitations of the analyses are explicitly stated. Underlying data are provided in supplements and in links to databases.

Additional comments

Line 67: ‘thermal stress’
Line 77-79: for clarity, suggested to rephrase to: “…16 distinct lineages, with Foraminifera Clade G, Clade Fr2, Clade Fr4, Clade I, and Clade J representing undescribed genera.”
Line 114-115: suggested edit: “…shore, with depths ranging from 1 to 10 m…”
Line 125: remove ‘sample’
Line 133: remove ‘frequently smaller ones’
Line 134: remove ‘had just been’
Line 136-138: move ‘via commercial courier service,’ to after ‘University’
Line 163: shouldn’t master mix be 1x not 0.8x in the final PCR reaction mix?
Line 176-179: suggest rephrasing for clarity “After quality checking, only 41 out of 72 samples were of sufficiently high quality for sequencing (see Table 2.). The low-quality of some libraries may be due to eDNA degradation during sample transport and extraction.”
Line 202: database version or date of access would be useful to add here
Line 236-237: not sure what the authors mean by “and please, see the relevant analyses code for additional information on data availability”
Line 240: omit “Nonparametric statistic of the”
Line 241-244: it is not clear here what method is being used to compare each parameter. It is suggested that authors split this section into multiple sentences to better convey each analysis.
Line 248: “DNA was successfully”
Line 254-255: suggest rephrasing to “Based on the total ASVs classified to taxon level 4 according to the SILVA database, it appears that unclassified Eukaryotes (43.35%) are the dominant taxa.”
Line 262: result > resulted
Line 285-286: please rephrase for clarity
Line 286 and 288: remove ‘sample’
Line 293: ‘it was’
Line 294: remove “much”
Line 297: “…relative abundance, it appears…”
Line 299: remove ‘analyses’
Line 310: please rephrase for clarity
Line 329-331: please rephrase for clarity
Line 333-334: please rephrase for clarity
Line 344: ‘researches’ > ‘research’
Line 346-347: ‘At least seven Symbiodiniaceae genera have been discovered…’
Line 344: isolated or just identified?
Line 344-356: please rephrase for clarity; this section can be made more concise as the main point is that different methods were used by various groups studying Symbiodiniaceae in the region, but they did not detect as many genera as the present study did
Line 361-363: please rephrase for clarity; not clear which studies are being referred to
Line 406-407: not sure that this sentence belongs here
Line 412: ‘Some of these can (re-) establish stable host-algal mutualisms (transient free-living), while others are true free-living Symbiodiniaceae, such as E. voratum’
Line 425: remove ‘were’
Line 429: ‘genetic material’
Line 429-431: please rephrase for clarity
Line 443: ‘A previous study…’
Line 444: eDNA in sediments?
Line 445: ‘After all,’ > ‘However,’
Line 896 (Figure 3 title): ‘Maximum likelihood phylogenetic tree based on V9 18S rRNA gene for Order…’
Line 918 (Figure 6 title): ‘…coral reefs…’
Line 998 (Table 4 title): ‘…present/absent analysis…’

Reviewer 2 ·

Basic reporting

Generally fine but see comments below.

Experimental design

As long as the lack of blanks is addressed, seems fine.

Validity of the findings

As long as the lack of blanks is addressed, seems fine.

Additional comments

This version is much improved; an overall good job of revision by the authors. I know the authors have many circumstances to deal with, and despite my comments below, I really want to see this work published and think with one more round of revision they should be able to reach the finish line. Please do not give up!

The English still needs a round of strict editing.

As well, some minor issues remain:
lines 77-79: wording awkward; rewrite.
lines 88-89: Reimer & Todd (2009) missing here, reported on Symbiodiniaceae from Singapore.
line 106: again, why focus only on coral here?
lines 114-116: past tense please. Also lines 262-263, 295, 329
line 121-122: give total n here to be very clear to readers.
line 197-199: the authors have not understood my comments about the "cutoff". By this, I mean arbitrarily choosing 97% and 99% as your thresholds. I still think the authors would be served better by checking into Symbiodiniaceae literature on the 18S marker and choosing the appropriate threshold for this family, and not relying on SILVA alone.
line 215: "These parameters setup will..." is awkward; reword.
line 307: You say here various organisms and yet your references seem to focus on scleractinians and corals - couldn't you cast the net wider?
line 329-337: this added paragraph is welcome, but is awkwardly worded. I would also say something like "lack of controls/blanks can XXXX, but in our study, we XXXXX, so we are confident in our results" or something similar. The wording comes across now as a bit ambiguous.
line 344: "researches" is uncountable, please correct.
line 370: Many members of Cladocopium - not all!

---

## Round 0.4 · accepted · Accept

Dear Authors,

I have now carefully assessed your final responses and revised manuscript and I am delighted to accept it for publication in PeerJ.

I was glad to see that care was taken to incorporate the final corrections made by the reviewers, and that the manuscript went through a solid proofreading of the English - thank you!

I did review the manuscript myself, and have found a number of minor corrections/considerations that I would like you to incorporate at the proofs stage - please note my comments and suggestions in the attached 'Editor Annotated' version of the manuscript.

I wanted to also pay my respect to Dr Hawis Madduppa who will be dearly missed. Thank you to all co-authors for representing Hawis and for your continued effort in carrying his legacy.

With warm regards,
Xavier